# Attitudes of Dental Students towards the Prescription of Antibiotics during Endodontic Treatment

**DOI:** 10.3390/antibiotics13100913

**Published:** 2024-09-24

**Authors:** Lauzan haj Khalaf, Salma Kabbaj, Babacar Toure

**Affiliations:** Department of Conservative Dentistry and Endodontics, Faculty of Dental Medicine, College of Health Sciences, International University of Rabat, Rabat 11103, Morocco; lauzan.haj.khalaf@uir.ac.ma (L.h.K.); salma.kabbaj@uir.ac.ma (S.K.)

**Keywords:** antibiotics, apical periodontitis, prescription, pulp pathologies, students

## Abstract

Aim: This study aimed to evaluate the antibiotic-prescribing attitudes of dental students during the management of endodontic infections. Materials and methods: This study was conducted in the five faculties of dental medicine in Morocco. A self-administered questionnaire was used and completed online. This questionnaire has three parts: the first includes the socio-demographic data, the second is related to the types of antibiotics prescribed, and the final addresses clinical scenarios in which antibiotics are prescribed. Data were analyzed with Jamovi, and X_2_ and ANOVA tests were performed. Results: Three hundred and sixty-five students participated in this study. The average duration of antibiotic therapy was 5.87 ± 1.45. Of all the students, 83.8% prescribe amoxicillin first. For patients with penicillin allergy, clindamycin was the most prescribed, amounting to 53.9%. Antibiotics are prescribed for all pulp and periapical pathologies. For acute pulpitis and acute apical periodontitis, a statistically significant difference between the different faculties was noted (*p* = 0.03). Regarding apical abscesses, antibiotic prescription was more frequent at the public faculty of Casablanca, corresponding to 92.8%. (*p* = 0.02). Conclusion: It appears from this study that there is a need for faculties to develop innovative teaching models to improve students’ level of knowledge on antibiotics and their indications in endodontics.

## 1. Introduction

According to the European Society of Endodontics, the use of antibiotics is recommended when an infection is persistent or systemic. Thus, for rational use, consensus conferences were organized to codify their indications [1]. Antibiotics are used in the following pathological situations: acute apical abscesses in medically compromised patients; patients at risk; and acute apical abscess with systemic involvement, i.e., localized swellings, fever > 38°, asthenia, insomnia, malaise, lymphadenopathy, trismus, progressive infections, rapid onset of severe infections, and dental reimplantation after expulsion [2]. Consequently, the prescription of antibiotics is not systematic in endodontics. Therefore, their inappropriate use can contribute to the occurrence of antibiotic resistance. The Lancet Infectious Diseases Commission has published a series of articles that sound the alarm on antibiotic resistance and encourage the community of practitioners to be more cautious when prescribing [3,4]. In this context, several medical professionals have started studies on the use of antibiotics. In endodontics, these studies have focused on general practitioners in several countries [5,6,7,8]. The results of these studies showed that very few practitioners use antibiotics appropriately in the management of pulpal and periapical pathologies [9,10,11,12]. An analysis of the literature shows that very few studies have focused on the attitudes and practices of students in clinical training. On this topic, according to the drug regimen, Mohanty et al. [13] reported that 30.85% of postgraduate students of endodontics prescribed pretreatment medication; in comparison, 62.40% of the analyzed endodontists preferred medication for both. European training program guidelines state that graduates should be adequately trained in the basic and clinical science of endodontics. They specify that students must not only have knowledge of the microbiology of pathologies of endodontic origin but also be competent enough to manage these pathologies. This management integrates infection control, pharmacology, and endodontic therapy.

In Morocco, the book of educational standards for dental studies mentions that dental students should know how to use antibiotics in managing microbial infections, the mechanisms of action, and the issues related to antibiotic resistance. They must also be able to perform simple root canal treatments. In clinical programs, students learn to apply their knowledge of antibiotics to manage infectious dental diseases. At the end of these courses, they must know the indications and contraindications for the use of antibiotics in endodontics.

Despite these guidelines in training programs, studies carried out among students in training in Spain reveal a lack of knowledge and inappropriate treatment regimens. They also show a serious need to improve knowledge in regard to the prescription of antibiotics [14].

In Morocco, no study has analyzed the practices of students concerning the use of antibiotics during endodontic treatments.

The objective of the present study was to assess the level of knowledge and prescription attitudes of students in the management of pulpal and periapical pathologies.

## 2. Materials and Methods

This study is based on a cross-sectional descriptive survey conducted from November 2021 to February 2022.

The study was approved by the Ethics Committee of the University Clinic of Dental Medicine of the International Faculty of the International University of Rabat (CUMD/FIMD 03/22). This study was conducted on all dental students from the 4th to the 6th year enrolled at 5 faculties of dental medicine in Morocco: The International University of Rabat (UIR), the Abulcassis University of Health Sciences (UIASS), the Mohammed 6th University of Health Science (UM6SS), the public faculty of Dental Medicine of Rabat (FMPR), and the public faculty of Dental Medicine Casablanca (FMPC).

The sole prerequisite for participation was to be a dental student in the 4th to 6th year of undergraduate studies. The questions were based on those asked in previous surveys developed in several counties [5,6,7,8].

A self-administered questionnaire (Table 1) based on models from previous studies was used [6,7,8,9,10,11,12]. This questionnaire has three parts.

The first part includes the socio-demographic data of the students: gender, level of study, and dental medicine faculty of origin. The second part is related to the types of antibiotics prescribed during endodontic treatment of an adult patient with or without an allergy to penicillin and the duration of the prescription of these antibiotics.

The third and final part addresses clinical scenarios in which antibiotics are routinely prescribed. The students who took part did so on a voluntary basis, without any form of compensation, and in an anonymous manner.

This questionnaire was sent online via email; a reminder was sent every 15 days. Questionnaires received after February 2024 were not included in the statistical analysis.

Data were collected and analyzed with Jamovi version 1.8.1 (software). Chi-square test and ANOVA tests were performed to compare qualitative and quantitative variables. The significance level was set at *p* < 0.05.

## 3. Results

In total, 365 students responded to the survey; 71.9% were female, and 28.1% were male, which yields a sex ratio of 2.49 in favor of women. Sixth-year students were more represented, with a percentage of 48% (169 students), than fifth-year students, constituting 28.7% of the sample (101 students), and, finally, the cohort comprised 82 students, representing 23.3% of the total number of students in their fourth year. The International University of Rabat (UIR) was the faculty that participated the most in this study, with a percentage of 39.9% (144 students); then, in second place was the Abulcassis University of Health Sciences (UIASS), with a percentage of 19.9% (72 students), and the Mohammed 6th University of Health Science (UM6SS), with a percentage more or less equivalent to that of Abulcassis, i.e., 19.7% (71 students). The public faculty of Dental Medicine of Rabat (FMPR) accounted for a percentage of 11.1% (40 students), and Casablanca (FMPC) had a percentage of 7.8% (28 students) (Figure 1).

The average duration was 5.87 ± 1.45 days, with a maximum of 7 days for 56.5% of the students and a minimum of 3 days for 14% of the students. No statistical difference was found between the different universities (*p* > 0.05).

### 3.1. For a Patient without a Penicillin Allergy

Most of the students, 83.8%, prescribe amoxicillin as their first intention, specifically the 1 g form of amoxicillin, which is the most prescribed. Amoxicillin associated with clavulanic acid was prescribed by 9.2%, azithromycin was proposed by 2.5% of students, and clindamycin was chosen by 1.4%. Concerning the prescription of Metronidazole associated with Spiramycin, we note a prescription percentage of 3.1%.

### 3.2. For a Patient with a Penicillin Allergy

Concerning patients with a penicillin allergy, clindamycin was the most prescribed molecule (53.9%), followed by azithromycin (20.9%). Erythromycin was indicated in 7.8% of cases, and metronidazole combined with spiramycin was indicated in 17.4%.

### 3.3. Endodontic Pathologies with Antibiotic Prescription

Antibiotics are prescribed for all pulp and periapical pathologies, whether acute or chronic (Table 2). For acute reversible and irreversible pulpitis, antibiotic prescriptions were more indicated by fourth-year students than by students in higher years. In cases of acute reversible pulpitis (21), 2% of fourth-year students prescribed antibiotics, compared to 9.9% of fifth-year students and 2.9% in their sixth year. This difference was statistically significant (*p* = 0.01).

For periapical pathologies, antibiotics were more indicated for acute apical abscesses, with percentages reaching 76.6%. No statistically significant difference was noted between the study levels. Regarding pain following endodontic treatment, antibiotic prescriptions were recommended by 20% of fourth-year students, 15.8% of fifth-year students, and 14.8% of sixth-year students; this difference was not statistically significant. However, a statistically significant difference was observed regarding the prescription of antibiotics in cases of pulp necrosis and endodontic retreatment. Students in their fourth year were found to have a higher rate of prescription than their counterparts at other levels.

The analysis of the results according to the different universities shows statistically significant differences concerning the prescription of antibiotics for acute apical abscesses, acute apical periodontitis, and endodontic surgery (Table 3).

The percentage of students who prescribed antibiotics in cases of apical periodontitis was significantly high at the Public Faculty of Casablanca (FMPC), amounting to 28.5%, compared to the percentages at other faculties (UM6SS, 18.5%; FMPR, 17.5%; and UIR, 9.8%, with *p* = 0.03).

For acute apical abscesses, the prescription of antibiotics was also higher among the students of the Public Faculty of Casablanca (92.8%) than at the other faculties (*p* = 0.02).

## 4. Discussion

This study deals with the knowledge and practices of students concerning the use of antibiotics during the management of pulpal and periapical pathologies. The questionnaire used in this study was proposed in previous studies carried out in several counties [12,14,15].

The sample consisted of 365 students from five faculties of dentistry in Morocco. Three of these faculties are in Rabat, namely, the International University of Rabat (UIR), Abulcassis University of Health Science (UIASS), and the Public University of Rabat (FMPR), and two of these faculties are in Casablanca, i.e., the Mohammed 6 University of Health Science (UM6SS) and the Public Faculty of Casablanca (FMPC).

Our sample is fairly representative because in the average percentage of participation recorded in similar studies [15,16], the number of women was higher (71.9%). This difference was also noted in Spain, which shows a feminization of the profession. The noted prescription period of 5.87 ± 1.45 days is in line with the recommendations because endodontic infections always regress within 3 to 7 days if the infectious cause is eliminated [17]. For patients without penicillin allergy, amoxicillin is the molecule of choice for students, with 83.8% selecting this antibiotic, followed by the combination of amoxicillin and clavulanic acid, at 9.2%. These results corroborate the data found in the study by Khaloufi et al., carried out among practitioners in Northern Morocco [15]. Furthermore, these results are comparable to those observed in other European countries [16] and Africa [18].

In India, practitioners also prefer amoxicillin as their first choice, followed by oxofloxacin. Amoxicillin is an effective antibiotic for the germs implicated in periapical pathologies.

However, in the event of inefficiency linked to the production of β-lactamase, the combination of amoxicillin with clavulanic acid can be proposed. According to scientific societies, this combination should be considered a second-line treatment in the event of failure using amoxicillin or for patients with established immunity [1]. For patients with a penicillin allergy, clindamycin was the most prescribed molecule (53.9%), followed by azithromycin (20.9%). Erythromycin was indicated in 7.8% of cases, and metronidazole combined with spiramycin was indicated in 17.4%. The study by Bolfini et al. showed results identical to this study, where clindamycin was the most prescribed antibiotic, at 33%. Rodriguez Nunez et al. [19] also reported clindamycin to be in first place, at 69%, followed by azithromycin, at 29.2%. In contrast, Al Khuzaei et al. [7], in a study carried out on dental surgeons, found different results, with azithromycin being at the top of the line, with 63.2%. Metronidazole occupies third place in this study. It is an anti-infective effective against bacteria with black pigmentation, but it is less effective on aerobes or facultative anaerobes, which makes it more advisable for it to be combined with another antibiotic such as amoxicillin or spiramycin. Its combination with amoxicillin should not be systematic but dictated by the evolution of the pathology; if there is no favorable evolution two to three days after the prescription of amoxicillin alone, metronidazole can be added to amoxicillin. In Spain, 99% of students chose clindamycin. The same study carried out among endodontists and general practitioners found different results (63% and 65% for clindamycin, respectively).

Regarding the indications of antibiotics as an adjuvant to endodontic treatment, several studies carried out in different countries show a lack of knowledge on indications and the inappropriate use of antibiotics. In this study, the percentage of students who prescribe antibiotics is relatively high. The results show that 9.1% of students prescribe antibiotics for acute pulpitis. A similar study carried out among students in Spain showed higher results for pulpitis (29%) and irreversible pulpitis with symptomatic apical periodontitis and moderate/severe symptoms (63%) [14]. However, these pathologies are dominated by pulpal inflammation; no trace of infection is noted in the pulp. Their management, therefore, does not require the use of antibiotics.

For chronic apical periodontitis, 15.4% of students prescribe antibiotics. These results are similar to those found in Spain (16%) [14]. This decrease in percentage could be mainly related to the chronic asymptomatic nature of these cases. However, compared to the results obtained by practitioners, the percentages are higher, with 31% in Spain and 51.2% in Morocco [15].

These results show the need for the continuous training of general practitioners.

In the case of retreatment, Moroccan students, like most dentists in India, prefer to prescribe antibiotics only in specific cases [20,21]. Otherwise, the use of a solvent during root canal retreatment does not cause any significant difference in the post-operative pain levels or medication intake for the retrieval of Gutta-percha [22].

Concerning postoperative pain, antibiotics were prescribed by dental students with a rate ranging from 7.5% to 22.8% (according to dental school). Therefore, Jose et al. [23] concluded that oral consumption of corticosteroids is a better analgesic in this case.

For acute apical abscesses, 76.6% of students prescribe antibiotics. In Spain [5,19], it is almost systematic, at 90%. Indeed, the prescription of an antibiotic should only be indicated in the presence of associated general signs [1,24].

## 5. Conclusions

The results of the present study show that despite the existence of pharmacology and endodontics modules in study curricula, the various faculties of dentistry should integrate new teaching methods to improve students’ knowledge of antibiotics and their indications in endodontics. They must develop innovative interactive teaching approaches based on real cases, electronic educational tools offering access to precise information, and standardized educational materials for prudent antibiotic therapy. The curriculum for dental studies should place more emphasis on prescription and the teaching of good practices.

## Figures and Tables

**Figure 1 antibiotics-13-00913-f001:**
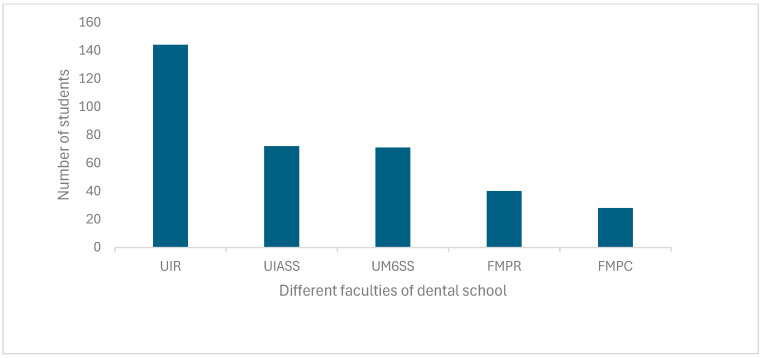
Distribution of the students according to the faculties of dental medicine. UIR, International University of Rabat; UIASS, Abulcassis University of Health Sciences; UM6SS, Mohammed 6th University of Health Sciences; FMPR, Public faculty of Dental Medicine of Rabat; FMPC, Public faculty of Dental Medicine of Casablanca.

**Table 1 antibiotics-13-00913-t001:** Antibiotic use in endodontic infections questionnaire administered to dental students’ undergraduates in Morocco.

Gender	Male □	Female □	
Undergraduate Year	4th□	5th□	6th□
Faculty of Dental Medicine: UIR □ UIASS □ UM6SS □ FMPC □ FMPR □
(1) When systemic antibiotics are indicated, which antibiotic would you choose for the treatment of an endodontic infection in an adult, healthy patient with no medical allergies? (choose one answer):
Amoxicillin	500 mg □	1 g □	
Amoxicillin + Clavulanic Acid	1 g\125 mg	_ □	
Azithromycin	250 mg □		
Clarithromycin	500 mg □		
Clindamycin	300 mg □		
Erythromycin Metronidazol + Spiramycin	500 mg □ _ □		
Other 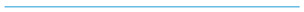
(2) For how many days would you prescribe an antibiotic treatment? 
(3) When systemic antibiotics are indicated, which antibiotic would you choose for the treatment of an endodontic infection in an adult, healthy patient with allergy to penicillin? (choose one answer):
Azithromycin 250 mg □ _ 500 mg
Clarithromycin 500 mg □ __ □
Clindamycin 300 mg
Erythromycin _ □ Metronidazole + spiramycin _ □
Other 
(4) In which of the following situations antibiotics are indicated?
Reversible pulpitis □
Irreversible acute pulpitis □
Apical acute periodontitis □
Chronic apical periodontitis without sinus tract □
Chronic apical periodontitis with sinus tract □
Acute apical abscess □
Pulp necrosis □
Endodontic retreatment □
Endodontic surgery □
Post-operative pain □

**Table 2 antibiotics-13-00913-t002:** Distribution of students who prescribe antibiotics according to endodontic pathologies.

N = Number of Prescriptions	4th	5th	6th	
% = percentage	N (%)	N (%)	N (%)	*p*
Reversible acute pulpitis	17 (21.2)	10 (9.9)	5 (2.9)	0.01 *
Irreversible acute pulpitis	14 (17.5)	8 (7.9)	11 (6.5)	0.06
Acute apical periodontitis	11 (13.7)	20 (19.8)	31 (18.4)	0.49
Chronic apical periodontitis	13 (16.2)	15 (14.8)	25 (14.8)	0.9
Apical periodontitis with fistula	32 (40)	43 (42.5)	80 (47.6)	0.67
Acute apical abscess	53 (66.2)	74 (73.2)	134 (79.7)	0.13
Pulp necrosis	11(13.7)	6 (5.9)	5 (2.9)	0.005 *
Endodontic retreatment	11 (13.7)	4 (3.6)	10 (5.9)	0.03 *
Endodontic surgery	31 (38.7)	48 (47.5)	88 (52.3)	0.11
Post op pain	16 (20)	16 (15.8)	25 (14.8)	0.47

* significant; N = number of prescriptions; % = percentage.

**Table 3 antibiotics-13-00913-t003:** Distribution of students who prescribe antibiotics for endodontic pathologies according to university.

	UIR N = 142	UIASS N = 72	UM6SS N = 70	FMPC N = 28	FMPR N = 40	*p*
	N (%)	N (%)	N (%)	N (%)	N (%)	
Reversible pulpitis	20 (14)	2 (2.7)	7 (10)	0	3 (7.5)	0.32
Acute pulpitis irreversible	11 (7.7)	7 (9.7)	12 (17.1)	0	1 (2.5)	0.47
Acute apical periodontitis	14 (9.8)	12 (16.6)	13 (18.5)	8 (28.5)	7 (17.5)	0.03 *
Apical periodontitis with fistula	58 (40.8)	30 (41.6)	38 (54.2)	18 (64.2)	16 (40)	0.11
Acute apical abscess	95 (66.9)	56 (77.7)	55 (78.5)	26 (92.8)	33 (82.5)	0.02 *
Phoenix abscess	53 (37.3)	28 (38.8)	30 (42.8)	18 (64.2)	19 (47.5)	0.03 *
Pulp necrosis	11 (7.7)	2 (2.7)	5 (7.1)	2 (7.1)	1 (2.5)	0.43
Endodontic retreatment	8 (5.6)	7 (9.7)	6 (8.5)	2 (7.1)	2 (5%)	0.8
Endodontic surgery:	52 (36.6)	41 (56.9)	44 (62.8)	16 (51.1)	15 (37.5)	0.004 *
Post op pain	20 (14)	13 (18)	16 (22.8)	6 (21.4)	3 (7.5)	0.6

* significant; N = number of prescriptions; % = percentage. UIR, International University of Rabat; UIASS, Abulcassis University of Health Sciences; UM6SS, Mohammed 6th University of Health; FMPR, Public faculty of Dental Medicine of Rabat; FMPC, Public faculty of Dental Medicine of Casablanca.

## Data Availability

Data is contained within the article.

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
