# Peer review of "Attitudes of Dental Students towards the Prescription of Antibiotics during Endodontic Treatment"

_antibiotics, 2024, doi:10.3390/antibiotics13100913_

Round 1
Reviewer 1 Report
Comments and Suggestions for Authors
- The author must approve the references, since not all of them have the same style.
- In the summary in the results part it is not understood, the author refers: Three sixty-five students, what number does he mean? Later in the same paragraph amoxicillin is misspelled, the author put "Amoxcilin", in the methodology there are also grammatical errors
- In the summary in the results part it is not understood, the author refers: Three sixty-five students, what is the number you mean? Later in the same paragraph amoxicillin is misspelled, the author put "Amoxcilin", in the methodology there are also grammatical errors, what type of survey they used or what type of instrument they used to measure the results (in relation to the title and the development of the article, they mention the attitude of the doctors, however they are measuring the type of antibiotic they prescribe)
- The topic is not new, it is known that in recent and future years antibacterial resistance will be incredibly high, with this type of empirical articles it only shows that antibiotics continue to be prescribed without justification alike.
Comments on the Quality of English LanguageIn the summary in the results part it is not understood, the author refers: Three sixty-five students, what is the number you mean? Later in the same paragraph amoxicillin is misspelled, the author put "Amoxcilin", in the methodology there are also grammatical errors, what type of survey they used or what type of instrument they used to measure the results (in relation to the title and the development of the article, they mention the attitude of the doctors, however they are measuring the type of antibiotic they prescribe)
Author Response
Reviewer(s)' Comments to the Author:
(I) We noticed that the main text of your manuscript is quite brief, which may mean that the experiment, research background, future research directions, or possible
applications of the research are not described in enough detail.
We want to inform you that all the necessary explanations are now included in the main text.
(II) Please consider the following points in your revisions: adding full experimental details, presenting completely all the results, and describing a comprehensive background to the research in the introduction section, including more references, providing additional figures and tables, and writing a conclusion section to summarize your research.
it's already done.
(III) Please check that all references are relevant to the contents of the manuscript.
- it's done.
(IV) Any revisions to the manuscript should be highlighted, such that any
editors and reviewers can easily review changes.
All changes have been highlighted.
(V) Please provide a short cover letter detailing your changes to the editors’ and referees’ approval.
It’s done.
(VI) If the reviewer(s) recommended references, please critically analyze
them to ensure that their inclusion would enhance your manuscript. If you believe these references are unnecessary, you should not include them.
The reviewer(s) didn’t recommend references
(VII) If one of the referees has suggested that your manuscript should undergo extensive English revisions, please address this issue during revision.
It’s done
Reviewer 2 Report
Comments and Suggestions for Authors
Dear authors,
The article deals with a questionnaire applied to dental students regarding the use of antibiotics in endodontics at 5 universities.
Here are some suggestions:
1. Put information about the questionnaire in the abstract.
2. In the figures and tables, include the meaning of the university acronyms.
3. Cite the universities and their acronyms in the methodology or results.
Author Response
The article deals with a questionnaire applied to dental students regarding the use of antibiotics in endodontics at 5 universities.
Here are some suggestions:
- Put information about the questionnaire in the abstract.
Information about the questionnaire has been added to the abstract
- In the figures and tables, include the meaning of the university acronyms.
The meaning of the university acronyms has been added to the figures and tables
- Cite the universities and their acronyms in the methodology or results.
The universities and their acronyms were cited in the methodology and results
Reviewer 3 Report
Comments and Suggestions for Authors
The aim of this paper was to evaluate the antibiotic-prescribing attitudes of dental students during the management of pulpal sand periapical pathologies.
Its main contribution is that adds information about the level of knowledge of students from different five faculties of dental medicine in Morocco, to the few existing information in the literature about dental students from another countries (Spain, Brazil, Saudi Arabia, Iran)
The manuscript is clear, and presented in a well-structured manner and does fit the journal scope, but I have some observation to make:
1. I don't know the Moroccan legislation on the students' right to practice on human patients, but I hope that their answers to the questionnaire only refer to the level of theoretical knowledge and their opinion, not to the antibiotic treatment that they really gave to the patients. Normally, a student can only have an opinion on a treatment indication, but his activity must be supervised by an experienced teaching staff, which prevents the application of an incorrect treatment to the patient. It should also be specified in the manuscript what is the level of knowledge is for the 4th, 5th and 6th year students, because it is obvious that there is a difference between the three categories of students....2. According to Instructions for Authors, the abstract should be a total of about 200 words maximum (the abstract is 244 words)
3. About one half of the cited references are not mostly recent publications (within the last 5 years)
Author Response
The manuscript is clear, and presented in a well-structured manner and does fit the journal scope, but I have some observation to make:
- I don't know the Moroccan legislation on the students' right to practice on human patients, but I hope that their answers to the questionnaire only refer to the level of theoretical knowledge and their opinion, not to the antibiotic treatment that they really gave to the patients. Normally, a student can only have an opinion on a treatment indication, but his activity must be supervised by an experienced teaching staff, which prevents the application of an incorrect treatment to the patient. It should also be specified in the manuscript what is the level of knowledge is for the 4th, 5th and 6th year students, because it is obvious that there is a difference between the three categories of students....
Moroccan legislation allows students to treat patients in clinics and university hospitals under the permanent clinical supervision of their professors from their 4th to 6th year of university studies.
- According to Instructions for Authors, the abstract should be a total of about 200 words maximum (the abstract is 244 words)
The abstract was corrected
- About one half of the cited references are not mostly recent publications (within the last 5 years)
- one recent reference has been added